# Understanding Embodied Effects of Posture: A Qualitative Study

Karen Lin and Elizabeth Broadbent *

Department of Psychological Medicine, The University of Auckland, Auckland 1142, New Zealand
* Correspondence: e.broadbent@auckland.ac.nz

**Abstract:** Some evidence suggests body postures can elicit emotion. Compared to neutral postures, constrictive postures are associated with negative affect and low arousal, whereas expansive postures have shown mixed effects. Qualitative methods may allow insights into this phenomenon. We asked 15 participants (mean age 43 years) to adopt eight different expansive, constrictive, or neutral postures, drawn from previous power posing or postural studies. After a minute in each posture, participants were interviewed about how they felt and when they might adopt the posture in real life. Interviews were audio recorded and inductive thematic analysis conducted. Power poses were associated with power and confidence, but also aggression, arrogance, intimidation, and disrespect. The slumped posture was associated with sadness and low control, and the upright seated posture with being alert and apprehensive as well as formality. Neutral postures were associated with feeling relaxed and comfortable. These results suggest that expansive postures have mixed emotional effects, but are inappropriate in some contexts.

**Keywords:** embodied cognition; posture; qualitative; expansive; constrictive; dominant; submissive

## 1. Introduction

Psychological theories and contemporary experimental research support the idea that emotional states can influence physical posture and, conversely, physical postures can affect emotional states [1–5]. Theories of embodied cognition suggest that neuron populations within the modality-specific sensory, motor, and affective systems are highly interconnected and form a dual process relationship between physical and psychological states [4].

There have been two predominant approaches to research into the effects of body postures on psychological states, the first of which focuses on slumped versus upright sitting postures. Upright posture has been shown to increase pride and persistence on tasks [5,6]. Later research using this paradigm has shown that an upright posture helped to maintain self-esteem and positive mood compared to a slumped posture during a stressor [7]. In a similar study, an upright posture was shown to improve mood and reduce fatigue in people with mild–moderate depression [8].

The second, more controversial, approach emerged in the 2010s, and focused on power poses, in which a powerful person might pose to exert influence over another. Based on participants' imagined behaviours of a boss versus a subordinate in an advertising firm vignette, or a person who takes charge and dominates others in social situations (or not), expansiveness and openness were chosen as two key features of socially powerful poses [9]. Images of four specific poses that differed on these dimensions were rated as differing in power by observers [10]; and people placed in the high-power poses reported feeling more powerful, had higher tolerance for risk, and had lower cortisol and higher testosterone. These effects were interpreted as advantaged and adaptive. However, there has been much controversy about this study, due to several failed replication attempts and questions about the methods employed [11]. Furthermore, the effects of power poses may not all be positive, and there is evidence that they can increase dishonest and self-interested behaviours. For example, expansive postures increased the likelihood of stealing money, cheating on a test, and committing traffic violations [12].

To help resolve some of the controversies and determine which effects were valid, two recent meta-analyses investigated the effects of both upright postures and power poses on various independent variables. A systematic review by Elkjaer et al. [13] included 48 studies in a meta-analysis, which indicated a significant overall difference between expansive and contractive postures across all combined contexts and outcome measures (Hedge's $g$ = 0.40), and between contractive and neutral postures (Hedge's $g$ = 0.45), but there was no significant difference between expansive and neutral postures. However, the review was criticised for not being pre-registered, not publishing the data analysis and code, missing relevant studies, and not distinguishing between various kinds of body positions [14].

A second systematic review included 117 studies in the meta-analysis and found a similar notable overall effect size, $g$ = 0.35 [15]. Again, detailed analysis showed the effects were only significant for slumped and low power poses compared to control groups, and not for upright and high power postures compared to control. The authors note, however, that not enough studies have included control (neutral) body postures, so we cannot fully know yet whether the effects are driven by the high or low power positions. The authors also found that the effects were larger for upright versus slumped postural manipulations than for power pose manipulations, and for individualist versus collectivist cultures. Both systematic reviews found effects for self-reported and behavioural outcomes, but not for physiological outcomes. Nevertheless, fewer studies have been conducted on physiological outcomes.

It has been proposed that social power has two components (dominance and prestige), and that power poses display dominance, whereas upright postures display prestige [15]. Dominance is linked with strength, threat, and intimidation, and is associated with aggression, narcissism, and low agreeableness, whereas prestige is linked to skills, abilities and knowledge, respect, and pro-sociality [16]. Körner and colleagues suggest that more research is needed to investigate the effects of both kinds of postures on these outcomes, and to investigate the specific meanings associated with certain body positions. Most studies to date have been limited to self-report questionnaires of mood, power, or risk-taking behaviours, which means other thoughts or experiences may not have been adequately captured. Qualitative research with open ended interviews may reveal additional information that may help explain how and why body positions affect psychological states.

There have been five qualitative studies on posture to date, which have predominantly focused on Eastern mind-body practices in clinical practice. Narrative analysis from qualitative interviews found that upright qigong postures (a system of co-ordinated body postures, breath, and meditation) during a 12-week treatment helped breast cancer survivors reconnect their mind and body, lessen their post-surgical pain, and foster acceptance of and confidence in their bodies [17]. Similarly, using semi-standardised interviews, Ref.[18] revealed that a nine-week yoga treatment helped patients with chronic neck pain emotionally distance themselves from burdensome cognitions via a renewed sense of body awareness. Research examining posture in people requiring mobility assistance used phenomenological analysis and found that an upright position removed a sense of disability and reinstated a sense of belonging, optimism, and normality [19], as well as providing opportunities to connect to the world through a new sense of self [20]. Only one mixed-methods study examined the effects of high and low power poses and neutral postures, in a learning environment using questionnaires and self-narratives in male students; upright postures resulted in feeling more energetic, active, and self-confident, with a higher level self-esteem and concentration compared to closed postures [21]. Further qualitative research is needed in other contexts and with mixed gender samples to understand the effects of expansive and constrictive postures on individuals.

This study aimed to explore experiences of adopting specific power poses and upright and slumped postures compared to neutral postures using qualitative methods. The findings may help understand why there have been mixed findings to date, and help examine the validity of distinguishing between power poses as non-verbal expressions of dominance, and upright postures as non-verbal expressions of prestige. A secondary aim

was to examine the extent to which these postures are adopted within daily life to assess their contextual relevance. The results may inform both theory and methods.

## 2. Materials and Methods

### 2.1. Participants

**Sampling.** This study used a convenience sampling strategy. It has been previously recommended that a sample size of 10 to 20 interviews is required to reach data saturation [22]. This is consistent with previous qualitative studies that conducted interviews for data collection on embodiment effects of posture [18,20]. Therefore, based on previous qualitative embodiment studies and best practice guidelines within the literature, a total sample of 15 participants was recruited.

**Eligibility Criteria.** The inclusion criteria required all participants to be (1) at least 16 years of age, (2) able to stand and sit without issues or assistance, and (3) fluent in English. Participants with diagnosed motor functioning, movement, or muscle disorders or conditions were excluded from this study.

**Recruitment.** Participants were recruited using printed advertising posters across the University campus and through emails sent to a faculty email group between 18 May and 8 June 2022. Interested parties received a participant information sheet and were screened for eligibility. After participants were deemed eligible for the study, they received an email outlining the available days and interview timeslots to participate in the study.

**Study Participants.** Fifteen participants were recruited, consented, and interviewed (eight female and six male) at the University campus within its clinical centre. Participants were aged between 27 and 77 years (M = 43 years) and identified as New Zealand European, British, English, Chinese, Filipino, Indian, African, Brazilian/Hungarian, and Nepalese. The highest level of education completed ranged from secondary school to undergraduate and postgraduate university degrees. Countries that participants most identified with in terms of cultural affiliation included New Zealand, United Kingdom (England), India, Philippines, South Africa, Brazil, Taiwan, China, and Nepal. Table 1 outlines the demographic characteristics of the participants.

**Table 1.** Participant Demographic Characteristics.

| Participant Number | Gender | Age | Ethnic Group | Highest Level of Education Completed | Country of Cultural Affiliation |
|---|---|---|---|---|---|
| 01 | M | 38 | Filipino | Postgraduate university degree | Philippines |
| 02 | F | 27 | NZ European | Postgraduate university degree | New Zealand |
| 03 | M | 27 | Chinese | Postgraduate university degree | New Zealand |
| 04 | F | 59 | Indian | Postgraduate university degree | India |
| 05 | F | 27 | Indian/African | Postgraduate university degree | NZ/South Africa |
| 06 | F | 53 | NZ European | Secondary school | New Zealand |
| 07 | F | 53 | British | University or polytechnic diploma | England |
| 08 | F | 33 | Brazilian/Hungarian | Postgraduate university degree | Brazil |
| 09 | M | 56 | NZ European | Undergraduate university degree | United Kingdom |
| 10 | M | 76 | NZ European | Undergraduate university degree | New Zealand |
| 11 | F | 37 | Chinese | Postgraduate university degree | Taiwan |
| 12 | M | 34 | English | Technical or trade certificate | England |
| 13 | F | 48 | Chinese | Undergraduate university degree | China |
| 14 | M | 37 | Nepalese | Postgraduate university degree | Nepal |
| 15 | F | 36 | NZ European | Postgraduate university degree | New Zealand |

NZ = New Zealand; M = male; F = female.

### 2.2. Procedure

All participants gave written informed consent. Participants completed a questionnaire on gender, age, ethnic group, current education level, and the country that the participant most identified with. Participants were then shown an image of a posture to adopt. Each participant remained in the given posture for one minute uninterrupted, after which the experimenter asked the participant the following questions:

1. How does being in this posture make you feel?
2. How often would you adopt this posture in your daily life?
3. Have you seen other people in this posture?

   Participants remained in each posture as they were questioned. This process was repeated for eight different postures. There were three postures from Carney et al. [10] and one posture from Cuddy et al. [23]. The other four postures included seated slumped and upright postures from Nair et al. [7], and neutral seated and standing postures. These postures were chosen to represent typical postures from past studies (see Figure 1). We chose three high power poses because these were quite varied and distinctive (shown in Figure 1 as 1, 4, and 5), compared to the other postures.

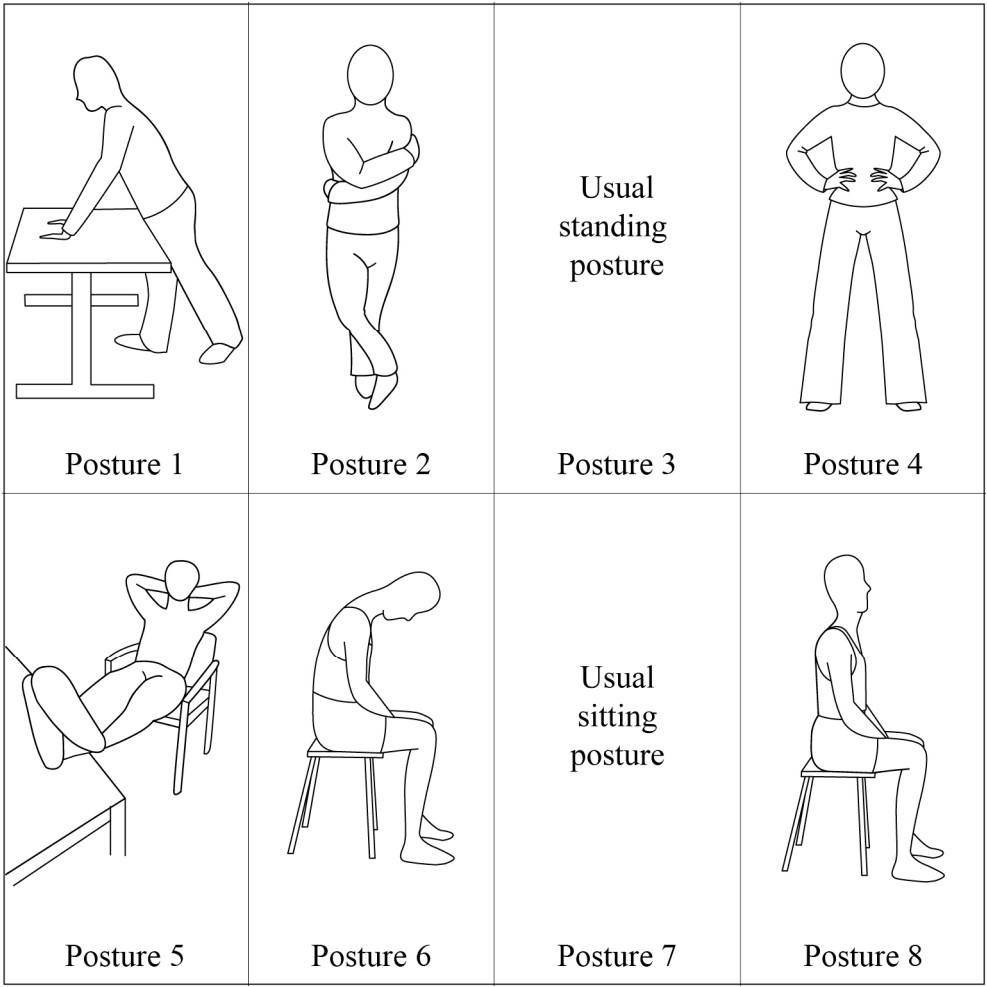

**Figure 1.** The Posture Manipulations. (1) Hands-spread-on-desk standing posture [10], (2) arms-and-legs-crossed standing posture [10], (3) usual standing posture, (4) hands-on-hips standing posture [23], (5) feet-on-desk sitting posture [10], (6) slumped sitting posture, (7) usual sitting posture, (8) upright sitting posture [7].

   Each interview took approximately 30 min and participants were given a NZ$20 gift voucher. The study was approved by the University Ethics Committee (AH24267) on 12 May 2022.

*2.3. Data Analysis*

   Interviews were transcribed verbatim into English by one researcher who then conducted thematic analysis on the data. Conducting inductive thematic analysis (TA) helped the researcher focus on the participants' standpoint through a detailed descriptive account of their experience for a bottom-up approach [22]. Following the general precepts of TA,

the researcher reviewed the data for initial coding. Codes were assigned to the data in a systematic manner across the entire data set and subsequently collated into potential themes. Data analysis was ongoing until new themes could no longer be identified, thus reaching data saturation. Data were refined to generate clear names and definitions for each theme. The thematic codes were checked by another researcher who had extensive research experience.

## 3. Results

Table 2 provides a summary of themes by postural set.

**Table 2.** Summary of Themes.

| Posture | Theme | Theme Description |
| --- | --- | --- |
| Hands-spread-on-desk standing posture | Power and intimidation<br>Attention and intimacy | Conveys power to another person.<br>Functions to increase intimacy. |
| Arms-and-legs-crossed standing posture | Comfort and protection<br>Unfamiliar and uncomfortable | Comforting and warm in a protective sense.<br>Functional for public or unfamiliar situations. |
| Usual standing posture | Familiar comfort<br>Multi-purpose function | A relaxing and common daily stance.<br>Suitable for a range of social and public situations. |
| Hands-on-hips standing posture | Power and confidence<br>Situational comfort | Conveys aggression and authority.<br>Situationally appropriate and sometimes habitual. |
| Feet-on-desk sitting posture | Perceived confidence<br>Embodied disrespect | Conveys the impression of relaxation and confidence.<br>Generates symbolic associations of disrespect. |
| Slumped-sitting posture | Low mood<br>Low control environments | Associated with sadness and worry.<br>Situations and environments likely to elicit sadness and worry. |
| Usual sitting posture | Default seat<br>Receptive | Normal daily sitting position.<br>Functions to receive information or anticipate further action. |
| Upright sitting posture | Alert and aware<br>Perceived formality | Enhances attention in anticipation of something to occur.<br>Reduced comfort of posture increases the perception of formality. |

### 3.1. Hands-Spread-on-Desk Standing Posture

#### 3.1.1. Power and Intimidation

Most participants felt that this posture conveyed power and an intent to exert influence over another person. Power and intimidation were characterised by invading another's personal space to make oneself "feel big" (P03, male, 27 years) and assert dominance.

Participants felt that towering or leaning over someone resembled belittling behaviour or interrogation tactics to conceal a lack of confidence deep down. They reported seeing similar depictions of intimidation and aggressive behaviour on television.

> *Typically people who are trying to um, you know um, trying to dominate other people or it's it's a sort of a um 'I'm in charge here' type of posture. (P10, male, 76 years)*

Most participants reported that they would infrequently adopt this posture and that it was unnatural and uncommon in daily life. Participants felt the stance was physically stressful and uncomfortable to hold.

> *[I've used this posture] near the end of March. I was very angry with the person. Um, yeah but otherwise [I] never use it. (P06, female, 53 years)*

#### 3.1.2. Attention and Intimacy

Some participants reported this posture as a functional stance to increase attention and intimacy in close relationships. One participant associated this posture with leaning in

to demonstrate something in a teaching role, while another participant associated it with getting closer to details in a work environment.

> . . . *kind of reminds me of like if I have a friend sitting on the table and I want to talk to them I could be like this to get a bit closer to talk . . . or a mum with a baby. (P08, female, 33 years)*

> *If I'm looking ar–looking at plans or architectural drawings around a table with a grou– group of colleagues or something . . . . (P12, male, 34 years)*

### 3.2. Arms-and-Legs-Crossed Standing Posture

3.2.1. Comfort and Protection

Participants described feeling secure and warm in this posture. According to participants, people may adopt this posture for physical warmth or if they are feeling unwell.

> *If I'm waiting for [a] bus that didn't come on time and I'm cold outside I might be like yeah, like um embracing myself and um, standing [on] one leg to conserve energy. (P11, female, 37 years)*

Participants reported this posture to be defensive, introspective and closed-off. They considered the crossed position of the arms and legs to be an act of protection but also acknowledged the stance to be unwelcoming and "falsely shut-off [to people]" (P09, male, 56 years). Thus, many participants reported generally avoiding this posture.

> *I also feel like it's like quite protective like I'm like hugging myself. Maybe like a comforting, protective type of posture. (P15, female, 36 years)*

> *Its—it's not warming and inviting to converse with people like this. (P12, male, 34 years)*

3.2.2. Unfamiliar and Uncomfortable

Participants reported feeling physically uncomfortable, unbalanced, and wobbly. According to participants, this posture might be adopted in social situations when people feel obliged to make chit-chat with unfamiliar people or in situations when people are waiting and impatient.

> *It's a very unstable position. (P01, male, 38 years)*

> *I was just thinking that if I'm leaning against a wall, it could also be like talking to someone that um [pause] you're not that familiar with so the conversation– in the conversation you're not so relaxed . . . in a situation that um, that person's the only one and I feel obliged to make chit-chat to. (P11, female, 37 years)*

### 3.3. Usual Standing Posture

Figure 2 shows how each participant stood when asked to stand in their usual posture.

3.3.1. Familiar Comfort

All participants reported their usual standing posture as a relaxed and common daily stance. They felt comfortable, content, and open to others approaching them in this posture.

> *Um, yeah, it makes me feel relaxed. It's sort of my standard pose. It just–like my body feels where it needs to be in the sense that legs naturally apart supporting my trunk . . . (P12, male, 34 years)*

3.3.2. Multi-Purpose Function

Participants acknowledged that their usual standing posture is suitable for a range of social and public situations. Participants thought this was a posture in which they could easily communicate with others and observe the world around them.

> *[This posture is] good it's uh it's good to . . . take in stuff. It's easy to interact with people, easy to communicate, easy to observe ((laughs)) so yeah. [I stand like this] very often. (P04, female, 59 years)*

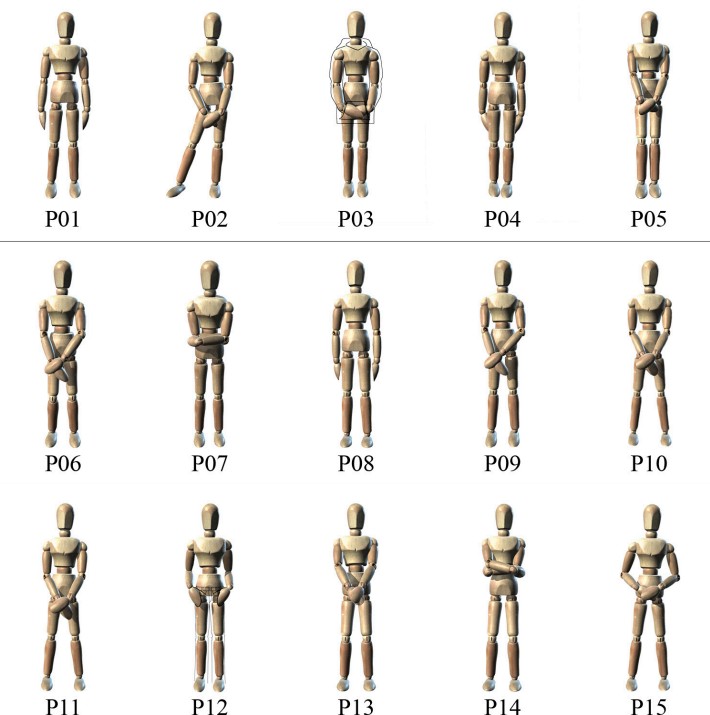

**Figure 2.** Usual Standing Posture of Each Participant. Note: added outlines indicate when hands were tucked into pockets.

*3.4. Hands-on-Hips Standing Posture*

3.4.1. Power and Confidence

Participants thought this posture conveyed aggression and authority. For some participants, positioning the hands on the hips indicated strength, authority, and a sense of exaggerated self-confidence often seen on television. Others saw this posture as confrontational and perhaps too confident because it gives the impression that a person "own[s] the space ... like they don't much care about what other people think" (P08, female, 33 years). Most participants reported rarely adopting this posture because it is potentially intimidating and rude.

*Maybe sometimes when you're very–I think the person is angry . . . . (P04, female, 59 years)*

*It's kind of assertive pose eh, I kind of see [a] teacher ... like [when] school kids are naughty and they kind of stand like that and telling someone off ... they're standing with their hands and they're kind of– you know, being grumpy. (P07, female, 53 years)*

3.4.2. Situational Comfort

Participants indicated that this posture might be appropriate for situations that require leadership or more direct instruction, such as in sports coaching, or formal conversations to "make you feel like people are listening to you more" (P07, female, 53 years).

*I play hockey and do some coaching so if I was talking about a drill maybe I would, you know, show the players how it works and then ... kind of stand to talk about any questions they might have or yeah, when I'm teaching if I was just talking about concepts. (P02, female, 27 years)*

*3.5. Feet-on-Desk Sitting Posture*

3.5.1. Perceived Confidence

All participants perceived this posture as relaxing and comfortable in informal situations. Several participants reported seeing this posture frequently adopted on television, in the movies or "on an American programme" (P13, female, 48 years) in which they perceived the posture to represent power and relaxation.

*Like . . . I feel like I'm setting myself up for holiday. (P12, male, 34 years)*

*Ah mostly people relaxing I guess, you know watching tv or . . . after work or something like this but I also feel like you see this in movies, you know like people in [a] corporate office, put their feet up on their desk in that big black armchair . . . . (P02, female, 27 years)*

Although many participants initially expressed a favourable impression toward the posture, upon physically adopting the posture, participants reported that it was painful to hold. Participants felt exposed and reported feeling psychological discomfort adopting this posture in work environments because it gave the impression of idleness.

*I think, yeah, I seen this posture like in movies or something but I never done it before but I thought that [it] looks powerful and relaxed? But I feel very awkward in this posture. (P11, female, 37 years)*

*I wouldn't do it at work . . . doesn't look like I'm working. (P09, male, 56 years)*

Participants reported thinking people in this posture were attempting to show dominance. They associated it with arrogance, a false sense of confidence, and a desire to appear in control. This posture was personally unfamiliar to many participants.

*It's the sort of thing you might see on an American programme where dominant bosses trying to . . . you know sort of . . . yeah, 'I'm not really interested in your opinion even though you're talking to me. I know I'm right and you're just not–of no importance'. (P06, female, 53 years)*

### 3.5.2. Embodied Disrespect

Several participants confirmed whether putting their feet on the table was acceptable behaviour before adopting the posture, while a few participants also removed their shoes before assuming the posture. Participants associated shoes with dirt and uncleanliness and thought that placing one's feet onto a table would be unhygienic, inconsiderate and rude.

*. . . in many parts of the world, um, to show the soles of your feet or your shoes to other people is–is an enormous insult . . . you don't put your feet on the table. (P10, male, 76 years)*

*. . . for me putting your feet on something is disrespectful . . . if it is a table which is meant for serving food then yeah, I don't. (P04, female, 59 years)*

### *3.6. Slumped-Sitting Posture*
### 3.6.1. Low Mood

Participants described this posture as heavy with sadness, fatigue, and worry. They associated this posture with regret and grief and reported that it reminded them of someone who was depressed, exhausted, deflated, and upset. Participants reported that the lack of eye contact with the world indicated either shameful avoidance or stress and pressure to find solutions after receiving bad news. Some participants recalled seeing children in this posture while hiding something or making a mistake. Participants felt the weight of the head over the shoulders resembled an inward position of someone feeling low, unhappy, and perhaps had their self-confidence knocked.

*I would associate this posture with, like, you're regretting something, you're sad uh . . . or you're like contemplating . . . like a mistake you've made. (P03, male, 27 years)*

*What I feel if I see people doing that I think they feel maybe, at least a little bit tired and also a little bit um [pause] unhappy. (P13, female, 48 years)*

Participants felt physically uncomfortable in this posture and reported that they would infrequently adopt it due to the unnatural position of the head and neck.

*This doesn't feel very normal to me, so..I don't think very often at all. (P05, female, 27 years)*

### 3.6.2. Low Control Environments

Participants reported that this posture was generally familiar but an unconscious position that people take in environments and circumstances when they are sad, or worried. Examples provided by participants included dementia centres, hospitals, outside a principal's office, visa offices, AA (alcoholics anonymous) meetings, and on the streets where people experience homelessness.

> . . . *makes me feel a bit sad, it reminds me of– like either, I dunno, like somebody at the hospital feeling sad or like a homeless person on the street. (P08, female, 33 years)*

> *Yeah, the doctors' and dentists' waiting rooms. Um [pause] you know, lot of people um . . . when I worked at the hospital, used to see a lot of people in waiting rooms and stuff just doing like this, you know. Child having an operation or something and thinking about stuff I suppose. (P09, male, 56 years)*

### 3.7. Usual Sitting Posture

Figure 3 shows how each participant sat when asked to sit in their usual posture.

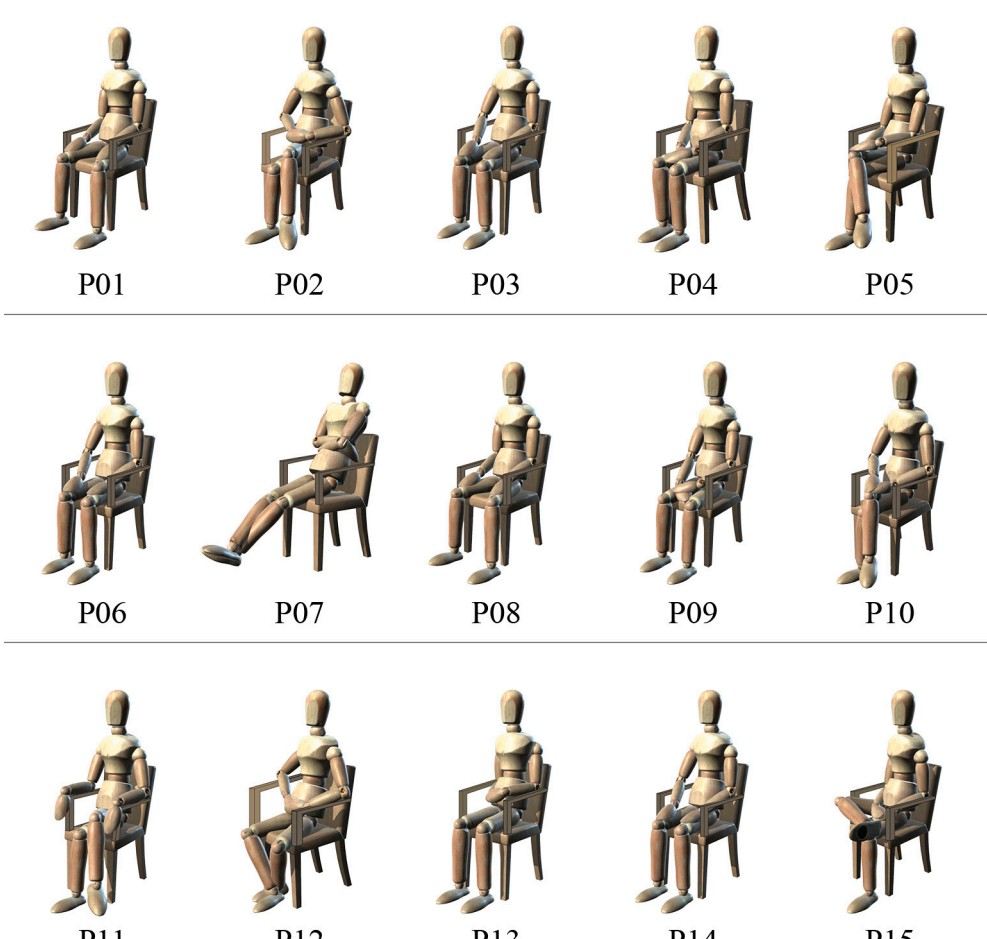

**Figure 3.** Usual Sitting Posture for Each Participant.

### 3.7.1. Default Seat

Across all participants, this posture was comfortable, relaxed, and frequently adopted. Participants reported this posture as a default sitting position for low-stress and casual situations. Participants felt they could maintain this posture for a longer duration and felt content.

> *Um I feel good. I feel relaxed. It's–yeah comfortable, quite happy, content sitting here. Yep. Can sit here for longer if I needed to. (P02, female, 27 years)*

### 3.7.2. Receptive

Participants indicated that they could easily receive information or anticipate further action in this posture. They described this posture as being commonly used when listening to others (e.g., in a meeting, during religious ceremonies) and waiting (e.g., for public transport). Participants felt that it was a suitable position for interacting with others in low-stress social situations, such as conversing with friends.

*Personally I think it's like a nice relaxed pose so you know if [people] were just like socialising or um . . . you know you've got your legs crossed cus that's how you're comfortable to sit at work or meetings or listening to someone talk. (P02, female, 27 years)*

### 3.8. Upright Sitting Posture

### 3.8.1. Alert and Aware

Participants experienced enhanced attention in this posture and described this posture as an apprehensive position that made them alert and more aware of their surroundings. Participants thought this posture helped keep them awake to appear more attentive and focused—as if they were trying to make a good impression or prepare for a serious conversation.

*Mm, it just feel like you going to do something. Like more like before your appointment of an interview. Um, like you want get yourself ready. You probably . . . not very nervous but maybe a little bit like nervous? (P13, female, 48 years)*

Participants reported that this posture is commonly used when a person is waiting or anticipating something. Examples included doctors waiting rooms, a job interview, and gymnasts at the Olympic games waiting to do their performance.

*. . . you feel like you're anticipating something. You're waiting or expectant. Um . . . you might feel a little nervous because you're . . . thinking um . . . might be, you know, if you're at a job interview sort of thing. You'd be running over . . . what you'd want to say and plan to say in your interview . . . (P06, female, 53 years)*

### 3.8.2. Enhanced Formality

Participants felt the reduced comfort of this posture also increased the perception of formality. One participant indicated that this posture might not be as frequently adopted because, generally, "people are more relaxed" (P08, female, 33 years). Participants reported that this posture felt "regimented and structured" (P12, male, 34 years) and that the reduced comfort was more conducive in foreign environments where personal space may be limited or when they needed to make a good first impression.

*I could be in a more sort of formal setting or . . . if I was out sitting somewhere, not– not on my own environment, sort of a bit of a foreign environment I'd probably sit a bit more bolt upright cus I wouldn't be as comfortable . . . . (P09, male, 56 years)*

*. . . when I'm . . . trying to cause a good impression so for example in a–a job interview I wouldn't sit like this [slouches] would sit like this [returns to assigned posture]. (P08, female, 33 years)*

## 4. Discussion

To our knowledge, this is the first qualitative study to examine the effects of adopting individual power poses and upright and slumped postures using semi-structured interviews and thematic analysis. Results support previously research on power posing; the hands-spread-on-desk standing posture was associated with power, dominance, and control [10,24,25]. Similarly, standing with hands on the hips and sitting with the feet on the desk were reported to represent power and confidence [10,23–26]. Participants' reports of sadness and worry associated with the slumped sitting position are consistent with earlier studies that found negatively-valenced emotions elicited from this posture—including significantly more unpleasantness and discomfort [27], increased fatigue and use of negative emotion and sadness words [7], negative thought processes [28], and reduced

self-confidence [5]. The arms-and-legs-crossed standing posture (a low power pose from previous studies) was reported as being closed off, defensive, and protective.

The upright sitting posture was reported to elicit higher alertness but also feelings of apprehension, formality and wanting to make a good impression. These findings contrast somewhat with previous results, which found upright sitting postures elicited more positively-valenced feelings of enthusiasm, excitement, higher self-esteem, reduced fear [7], and easier access to positive memories [28]. However, it should be remembered that previous studies often compared the sitting upright posture to a slumped position, which can influence the interpretation of the results. In addition, some of these studies included a stressor task during the postures, and sitting upright may be more appropriate when facing a threatening social context.

### 4.1. Theory

Overall, the findings are consistent with a dominance–prestige framework [15]. Dominance is defined by threatening and intimidating behaviours [16], and this was reflected in the comments from the participants in all three of the high-power poses. On the other hand, prestige is related to high status through knowledge and expertise. In this study, the slumped sitting posture was associated with a lack of self-confidence, having made a mistake, and shame, which all relate to low prestige. The upright sitting posture was associated with formal situations, being focused, and wanting to make a good impression, which are all linked to higher prestige.

In this study, participants described feelings related to affect, arousal, and dominance. The findings align with the PAD emotional state model [29], which has three dimensions—pleasure, arousal, and dominance. The pleasure dimension represents cognitive judgements of the quality of an affective experience (i.e., the pleasantness or unpleasantness of a posture). The arousal dimension represents the degree of excitement or stimulation one feels whilst in the posture. The dominance dimension describes the degree to which one feels they are in control over their life circumstances (i.e., power over one's situation). The PAD model has previously been used to study non-verbal communication in psychology [30,31]. However, this is not a complete model for examining the effects of posture, as research has shown effects on other outcomes, such as information processing and creativity [32,33].

### 4.2. Strengths and Limitations

Including neutral postures was a strength of the study. Several authors have recommended adding a neutral postural condition to allow the effect of manipulated posture to be ascribed to the appropriate expansive or constrictive condition in experimental studies. The neutral standing and sitting postures in this study helped to compare the degree to which the directed expansive or constrictive postures varied from participants' normal standing or sitting postures. These positions also helped provide a comparison for how people usually feel.

Several methodological aspects of the study may have influenced the results. First, since the participants saw and completed all the postures, their answers would have been influenced by having seen and/or experienced the other postures. Second, the participants experienced two aspects to the postures—their internal feelings when displaying the posture, and the social aspect of taking that stance relative to the interviewer. This helps to explain why the themes for each posture are contradictory in some cases. For example, sitting with the feet on the desk may feel relaxed and confident but also disrespectful to the interviewer. The question about whether the participants had seen others in the posture, may have caused a switch in perspective for the participants to be an observer rather than actor when answering this question. Therefore, participants experienced aspects of both roles, which is reflected in their responses.

A limitation of the study is that the findings may not generalise to other cultures. Participants affiliated with a range of cultures, and their experiences, are likely to have been influenced by their cultural upbringing. The degree to which power and dominance

are embodied as pleasurable emotions is culturally bound. Cultures nurture different views of what is desirable, meaningful, and appropriate in relation to power [34]. Previous studies investigating postural effects primarily originated from Western-oriented countries where feelings of power and domination are judged to be positively-valenced emotions associated with confidence and employment success [24,35]. Furthermore, Western cultures promote the notion that positive moods are more desirable than negative moods, and social desirability or appropriateness may be a significant component of emotional experience and expression [36]. In previous research, participants born in East Asian countries reported feeling less powerful than participants who were born in the USA when sitting with their feet on the desk [25]. This posture may violate cultural norms of humility, modesty, and restraint for Southeast Asians. The symbolic meaning of a posture can shape psychological experiences of participants from different cultures.

Interpretations of posture are also shaped by context. Previous studies [10,23] took place in business schools where certain behaviours and stances (e.g., hands spread on desk, hands on hips, and feet on the desk) may exude leadership and confidence. In contrast, this study took place in a medical school environment in which conveying authority and dominance via body posture may not have been as desirable. Thus, context-dependent effects need to be considered when interpreting these findings.

Age and gender influences should also be noted. While the mean age of participants within previous studies is mainly below 30 years, the older age of some of the participants within the current study may contribute to differences. Trait dominance is suggested to decrease with advancing age, meaning that as people age, they are less likely to feel they have control over their activities and life circumstances [37]. Moreover, there were some comments about gender appropriateness of some of the postures. Some individuals associated the arms and legs crossed standing posture, the feet on the desk sitting posture, or their own usual standing posture as a male stance. A male participant interpreted the hands-on-hips posture as the way women stand. This perception may reflect a generational orientation toward postural appropriateness for men and women, as the participants who made these comments were older.

Within a single posture, different feelings may be displayed by single segments of the body (e.g., the positions of the arms, legs, torso, and head). It would be interesting for future work to provide a finer grained analysis of which part of the body relates to each internal feeling.

*4.3. Clinical Implications*

This study was not performed with a clinical population, but the findings may still be informative for clinicians. Sad mood and low control were consistently reported by participants when in the sitting slumped posture, and, therefore, encouraging a more upright/neutral posture amongst depressed patients may help to improve mood, at least in the short term [8].

Further work could investigate posture and mood amongst populations with physical disabilities, for whom sitting or standing tall can be very extremely difficult. Assistive devices may help people with disabilities not only lead more active and functional lives, but could also have effects on their mood. Cases studies found that regular frame standing had positive psychological effects for people with severe multiple sclerosis, enhancing a sense of belonging and optimism by restoring life roles [19].

**5. Conclusions**

Overall, this qualitative study supports findings from quantitative research that expansive and open poses can increase feelings of power and confidence. However, some of these poses were experienced as aggressive, arrogant, disrespectful, and intimidating. The upright sitting posture was associated with feeling alert and prepared, but also feelings of apprehension, formality, and making a good impression. This mixture of negatively- and positively-valenced emotions may go some way towards explaining why systematic

reviews have found inconsistent effects of upright postures and high-power poses compared with neutral postures, which are experienced as comfortable and relaxed. In contrast, the slumped seated posture was consistently reported as being associated with sadness and fatigue, which aligns with meta-analytic findings that constrictive postures cause low mood compared with neutral or expansive postures. The dominant nature of power poses may cause some participants to feel aggressive, disrespectful, and arrogant, which makes them psychologically uncomfortable. Future research could explore whether scenarios in which social power is shown more positively towards subordinates (perhaps through sharing knowledge and skills), are associated with poses that still convey higher social rank, yet have more positive associations and psychological effects.

**Author Contributions:** Conceptualization, E.B. and K.L.; methodology, E.B. and K.L.; formal analysis, K.L.; investigation, K.L.; data curation, K.L.; writing—original draft preparation, K.L.; writing—review and editing, E.B.; visualization, K.L. and E.B.; supervision, E.B.; project administration, K.L. All authors have read and agreed to the published version of the manuscript.

**Funding:** This research received no external funding.

**Institutional Review Board Statement:** This study has been performed according to the APA ethical code This study obtained ethics approval from the Auckland Health Research Ethics Committee (AH24267) on 12 May 2022. This research was conducted following the principles within the Helsinki Declaration to ensure participants are respected, their health is safeguarded, and that they retain the right to make informed decisions throughout the study.

**Informed Consent Statement:** Informed consent was obtained from all subjects involved in the study.

**Data Availability Statement:** Participants of this study did not agree for their data to be shared publicly, so supporting data are not available.

**Conflicts of Interest:** The authors declare that this study has been conducted in the absence of any financial or commercial relationships that could be interpreted as potential conflicts of interest.

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
