# Peer review of "Understanding Embodied Effects of Posture: A Qualitative Study"

_psych, doi:10.3390/psych5020030_

Round 1

Reviewer 1 Report

The authors conducted a qualitative study using thematic analysis on expansive versus constrictive body positions. They asked 15 participants about their feelings after engaging in different power poses or upright / slumped postures. Overall, I found the manuscript well written, in line with common terminology, well structured, and the results are surprisingly strong in line with theoretical accounts on body positions. I agree with the authors that there seems to be no qualitative study on power poses so far and I see the necessity of this research to more fully understand the phenomenon of power poses. Therefore, I only have some minor comments, mainly pertaining to theory:

1)     I recommend not to over-interpret the difference between low power poses/slumped postures and control group participants. First, because of the criticism on the Elkjaer et al.-meta-analysis (https://doi.org/10.1177/1745691620984474) I have some serious doubts whether they are an appropriate reference for supporting such a notion – it might be good to briefly refer to that criticism. Further, I wrote with my colleagues, “Note that there is only a potentially small difference between power poses/upright postures and low-power poses/slumped postures in the set of studies that also has a control group due to the low number of studies (k = 14 and 11, respectively), g = 0.31, 95% CI [0.048, 0.580].” (see also the limitations section in the Psy Bull article). That is, so far, we do not fully know whether the effect is driven by the high power or the low power posing condition because not enough studies have included control groups to provide reliable evidence on this issue.

2)     You report the exact effect sizes of the two meta-analyses. However, the number of studies did not match with the effect sizes. Elkjaer et al. considered 73 studies relevant but only 48 studies were included in the meta-analysis (producing Hedge’s g of 0.40). The same applies to Körner et al. – indeed, we used 128 studies but only 117 studies were used for the high power posing/upright postures vs. low power posing/slumped postures comparison (resulting in g = 0.35). Please update the numbers accordingly.

3)     Line 96: I am not fully convinced that the qualitative study can address the issue of why a specific body position group (e.g., low power poses) should have stronger effects than another group (e.g., high power poses). However, I think the research can help to further examine the validity of distinguishing between power poses as nonverbal expressions of dominance and upright postures as nonverbal expressions of prestige. And indeed, the results seem to strongly support this difference.

4)     Three high power poses were used but only one body position each for the upright / slumped postures, the low power posing condition, and the neutral conditions. Was there any reason why three different high power poses have been used? I think high power poses are more interesting to compare against each other because they differ to a larger degree from each other than the other kinds of body positions but I still wondered why this decision was made. If there was no specific reason, I think this is also okay.

5)     It might help to provide specific names for the employed body positions (see e.g., Park et al., 2013; https://doi.org/10.1016/j.jesp.2013.06.001). For example, in the results section, you could replace “Posture 5” with “Expansive-feet-on-desk pose”. This would increase recognizability and readers would not have to go back to Figure 1.

6)     Discussion: I think participant’s reports after engaging in posture 2 do match with low power. Being defensive and protective matches the way a powerless (or better, submissive) individual perceives the world. Being constricted and perceiving threats is thus in line with theoretical accounts. So far, it reads a little bit as the reports about postures 2 are surprising but I think they are in line with what can be expected. Further, I think the brief interpretation of the findings can be somewhat stronger linked with the concepts of dominance (power poses) and prestige (upright postures).

7)     “Elkjaer et al. (2002) recommended adding a neutral postural condiction...” – this is actually recommended in almost every review on this topic. Writing, “Several authors recommended...” is more appropriate.

Overall, the authors conducted an interesting and important study and after addressing the minor points mentioned by me, I think the manuscript will be of value to the community.

Author Response

Thank you for inviting us to revise and resubmit our manuscript. We thank the reviewers for their thoughtful comments and have responded on a point by point basis, as indicated below.

The authors conducted a qualitative study using thematic analysis on expansive versus constrictive body positions. They asked 15 participants about their feelings after engaging in different power poses or upright / slumped postures. Overall, I found the manuscript well written, in line with common terminology, well structured, and the results are surprisingly strong in line with theoretical accounts on body positions. I agree with the authors that there seems to be no qualitative study on power poses so far and I see the necessity of this research to more fully understand the phenomenon of power poses. Therefore, I only have some minor comments, mainly pertaining to theory:

  • I recommend not to over-interpret the difference between low power poses/slumped postures and control group participants. First, because of the criticism on the Elkjaer et al.-meta-analysis (https://doi.org/10.1177/1745691620984474) I have some serious doubts whether they are an appropriate reference for supporting such a notion – it might be good to briefly refer to that criticism. Further, I wrote with my colleagues, “Note that there is only a potentially small difference between power poses/upright postures and low-power poses/slumped postures in the set of studies that also has a control group due to the low number of studies (k = 14 and 11, respectively), g = 0.31, 95% CI [0.048, 0.580].” (see also the limitations section in the Psy Bull article). That is, so far, we do not fully know whether the effect is driven by the high power or the low power posing condition because not enough studies have included control groups to provide reliable evidence on this issue. 

Thank you for your comments. We have now described the limitations of the Elkjaer paper and cited this paper. We have reduced the interpretation of the differences between low power poses/slumped group compared to neutral postures and added a comment that not enough papers have included neutral (control) postures to date to be able to make strong conclusions

  • You report the exact effect sizes of the two meta-analyses. However, the number of studies did not match with the effect sizes. Elkjaer et al. considered 73 studies relevant but only 48 studies were included in the meta-analysis (producing Hedge’s g of 0.40). The same applies to Körner et al. – indeed, we used 128 studies but only 117 studies were used for the high power posing/upright postures vs. low power posing/slumped postures comparison (resulting in g = 0.35). Please update the numbers accordingly.

Thank you for pointing this out – we have corrected the numbers accordingly.

  • Line 96: I am not fully convinced that the qualitative study can address the issue of why a specific body position group (e.g., low power poses) should have stronger effects than another group (e.g., high power poses). However, I think the research can help to further examine the validity of distinguishing between power poses as nonverbal expressions of dominance and upright postures as nonverbal expressions of prestige. And indeed, the results seem to strongly support this difference.

We have now updated these aims to examine the validity of the interpretations of postures as dominant (expansive) or prestigious (upright).

  • Three high power poses were used but only one body position each for the upright / slumped postures, the low power posing condition, and the neutral conditions. Was there any reason why three different high power poses have been used? I think high power poses are more interesting to compare against each other because they differ to a larger degree from each other than the other kinds of body positions but I still wondered why this decision was made. If there was no specific reason, I think this is also okay.

We have added some justification for these choices to the methods section.

  • It might help to provide specific names for the employed body positions (see e.g., Park et al., 2013; https://doi.org/10.1016/j.jesp.2013.06.001). For example, in the results section, you could replace “Posture 5” with “Expansive-feet-on-desk pose”. This would increase recognizability and readers would not have to go back to Figure 1.

We have renamed the postures according to their descriptions and referred to these throughout the text. We have renames these in the Table and Figure 1

  • Discussion: I think participant’s reports after engaging in posture 2 do match with low power. Being defensive and protective matches the way a powerless (or better, submissive) individual perceives the world. Being constricted and perceiving threats is thus in line with theoretical accounts. So far, it reads a little bit as the reports about postures 2 are surprising but I think they are in line with what can be expected. Further, I think the brief interpretation of the findings can be somewhat stronger linked with the concepts of dominance (power poses) and prestige (upright postures).

We have rephrased this, so that the results are not interpreted as surprising. We have added as sentence on the dominance-prestige framework to the discussion. 

7)     “Elkjaer et al. (2002) recommended adding a neutral postural condiction...” – this is actually recommended in almost every review on this topic. Writing, “Several authors recommended...” is more appropriate.

We have altered this phrasing in the discussion.

Overall, the authors conducted an interesting and important study and after addressing the minor points mentioned by me, I think the manuscript will be of value to the community.

Reviewer 2 Report

This study sought to understand the embodied effects of posture.

The study was very interesting and well-written.

Since this work is slightly outside my area of expertise, my only recommendations would be to include a brief discussion of the following.

-How might postural effects on attitudes/emotions vary across culture? A brief discussion thereof would make sense to contextualize this research on a global scale, recognizing that these are likely normative postural signals in certain cultures only. This should be stated as a limitation more explicitly.

-How will this work help improve the daily lives of individuals suffering from anxiety and depression? A discussion of the prescriptive implementation of suggested postural/behavioral changes would help clinicians, behavioralists, social workers, etc. see the tractability of your work.

-How can this work be adapted to physically disabled populations? A brief section on a discussion thereof would improve the breadth of populations to which your conclusions are actionable.

Author Response

Thank you for inviting us to revise and resubmit our manuscript. We thank the reviewers for their thoughtful comments and have responded on a point by point basis, as indicated below.

This study sought to understand the embodied effects of posture. The study was very interesting and well-written. 

Since this work is slightly outside my area of expertise, my only recommendations would be to include a brief discussion of the following.

-How might postural effects on attitudes/emotions vary across culture? A brief discussion thereof would make sense to contextualize this research on a global scale, recognizing that these are likely normative postural signals in certain cultures only. This should be stated as a limitation more explicitly.

We have stated more explicitly that culture is a limitation of the study. We have also added some further discussion of this point.

-How will this work help improve the daily lives of individuals suffering from anxiety and depression? A discussion of the prescriptive implementation of suggested postural/behavioral changes would help clinicians, behavioralists, social workers, etc. see the tractability of your work.

We have now added a short clinical implications section.

-How can this work be adapted to physically disabled populations? A brief section on a discussion thereof would improve the breadth of populations to which your conclusions are actionable.

We have added a short section on the applicability of this work to physically disabled populations. This is an interesting avenue for future research.

Reviewer 3 Report

The paper presents a qualitative study on the effects of posture. After having 15 participants take 8 postures and asking them how they feel in each of those poses, in which situations they would take it, and if they happened to see it in other people, the Authors cluster their answers in 16 themes, two for each posture.

The work is very interesting, both because it is the first case of a qualitative study on postures, and because it relies on a body version of the facial feedback hypothesis, proposed by Darwin and developed by others, according to which it is not only that an emotion triggers expressive behavior, but also a specific expressive behavior may trigger the corresponding emotion.

Yet, the paper suffers for some problems that should be, if not solved, at least mentioned and discussed by the Authors.

First, a methodological problem. The very fact that each of the 15 participants saw all 8 posture and answered questions about each of them makes the experimental design a within-subjects one, hence the answer about each posture is highly influenced by having seen the others.

A second more relevant problem is in the results and their discussion: the themes discovered as underlying the participants’ answers are almost in all cases contradictory if not opposite.

These contradictions are not due only to the cultural, gender, age variables supposed by the Authors in the discussion, but in a strong ambiguity in the study stemming from the fact that each posture has two aspects: on the one side, it is the expression of an internal feeling of the subject who displays it, but on the other it has a social objective of displaying the subject’s “stance” toward the Interlocutor.

Take the themes underlying posture 5: “Perceived confidence Embodied disrespect”

If Subject A displays  a high level of self-confidence by sitting as in Posture 5, this might be interpreted (and possibly be deliberately intended) as showing disrespect towards the Interlocutor.

Actually, in the questions to Participants,

1.        How does being in this posture make you feel?           

2.        How often would you adopt this posture in your daily life?      

3.        Have you seen other people in this posture?         

this double perspective, from the point of view of the Subject, and from the point of view of the Interlocutor, was not highlighted, but perhaps Participants, in answering question 3,  inadvertently switched from one to the other in their answers, that is, they reported not so much their own feeling when being in that posture but what (they thought) one generally thinks about people in that posture. This might account for the opposite valence of the two themes discovered in the Participants’ descriptions, the first generally positive because seen from one’s own perspective, the second often negative because judged from an Interlocutor’s perspective.

This is also the reason why the account proposed by the Authors in terms of Mehrabian’s (1996) PAD emotional state model results somewhat poor and not completely satisfying.

Since the end of last century, research on multimodality has clearly demonstrated that body expressive behavior are not only the carrier of emotions, but of many other internal states of subjects, including cognitive and relational stance, attitudes towards other people, and other contents. In this case one might say that each posture conveys both an internal feeling AND a social stance, and in some cases the stance, being viewed as by an Interlocutor, is judged in a different way.

A minor point:

At lines 384-385

The Authors show surprised by a specific result:

“Contrary to expectations, Posture Two (a low power pose from previous studies) was not specifically reported as low power, but rather as being closed off, defensive, and protective”. 

This is not so surprising, because what definitely in that posture gives the impression of closeness and defensiveness are the crossed arms, whereas the rest of the posture does in fact give the impression of low power. This depends on the fact that in a single posture different feelings may be displayed by single segments of the body; so a finer-grained analysis should be provided which sets the correspondences between each part of the body and each internal feeling. But since this was not intended in this study, it could only be pointed at as a possible objective of future work.

Finally, a minor issue is the mismatch between Figure 2 and Figure 3.

Figure 2 should in fact be n.3 and vice versa. In figure 2 the caption is right but the figures are wrong (they are sitting postures) and the other way around for Figure 3, where they are the standing postures instead of the sitting ones.

Author Response

Thank you for inviting us to revise and resubmit our manuscript. We thank the reviewers for their thoughtful comments and have responded on a point by point basis, as indicated below.

Comments and Suggestions for Authors

The paper presents a qualitative study on the effects of posture. After having 15 participants take 8 postures and asking them how they feel in each of those poses, in which situations they would take it, and if they happened to see it in other people, the Authors cluster their answers in 16 themes, two for each posture. 

The work is very interesting, both because it is the first case of a qualitative study on postures, and because it relies on a body version of the facial feedback hypothesis, proposed by Darwin and developed by others, according to which it is not only that an emotion triggers expressive behavior, but also a specific expressive behavior may trigger the corresponding emotion. 

Yet, the paper suffers for some problems that should be, if not solved, at least mentioned and discussed by the Authors. 

First, a methodological problem. The very fact that each of the 15 participants saw all 8 posture and answered questions about each of them makes the experimental design a within-subjects one, hence the answer about each posture is highly influenced by having seen the others. 

A second more relevant problem is in the results and their discussion: the themes discovered as underlying the participants’ answers are almost in all cases contradictory if not opposite.

These contradictions are not due only to the cultural, gender, age variables supposed by the Authors in the discussion, but in a strong ambiguity in the study stemming from the fact that each posture has two aspects: on the one side, it is the expression of an internal feeling of the subject who displays it, but on the other it has a social objective of displaying the subject’s “stance” toward the Interlocutor.

Take the themes underlying posture 5: “Perceived confidence Embodied disrespect”

If Subject A displays  a high level of self-confidence by sitting as in Posture 5, this might be interpreted (and possibly be deliberately intended) as showing disrespect towards the Interlocutor.

Actually, in the questions to Participants, 

  1. How does being in this posture make you feel?           
  2. How often would you adopt this posture in your daily life?      
  3. Have you seen other people in this posture?          

this double perspective, from the point of view of the Subject, and from the point of view of the Interlocutor, was not highlighted, but perhaps Participants, in answering question 3,  inadvertently switched from one to the other in their answers, that is, they reported not so much their own feeling when being in that posture but what (they thought) one generally thinks about people in that posture. This might account for the opposite valence of the two themes discovered in the Participants’ descriptions, the first generally positive because seen from one’s own perspective, the second often negative because judged from an Interlocutor’s perspective.

This is also the reason why the account proposed by the Authors in terms of Mehrabian’s (1996) PAD emotional state model results somewhat poor and not completely satisfying.

Since the end of last century, research on multimodality has clearly demonstrated that body expressive behavior are not only the carrier of emotions, but of many other internal states of subjects, including cognitive and relational stance, attitudes towards other people, and other contents. In this case one might say that each posture conveys both an internal feeling AND a social stance, and in some cases the stance, being viewed as by an Interlocutor, is judged in a different way. 

Thank you for raising these issues. We have added a paragraph to the discussion where we discuss the methodological issues described here. We have also added a sentence that the PAD model is not a complete explanation for the results.

A minor point:

At lines 384-385

The Authors show surprised by a specific result:

“Contrary to expectations, Posture Two (a low power pose from previous studies) was not specifically reported as low power, but rather as being closed off, defensive, and protective”. 

This is not so surprising, because what definitely in that posture gives the impression of closeness and defensiveness are the crossed arms, whereas the rest of the posture does in fact give the impression of low power. This depends on the fact that in a single posture different feelings may be displayed by single segments of the body; so a finer-grained analysis should be provided which sets the correspondences between each part of the body and each internal feeling. But since this was not intended in this study, it could only be pointed at as a possible objective of future work.

We have removed the part about the surprising nature of the results for posture two. We have added a point about a posture containing multiple single body parts to the discussion, and that future research could conduct a more fine grained analysis.

Finally, a minor issue is the mismatch between Figure 2 and Figure 3. 

Figure 2 should in fact be n.3 and vice versa. In figure 2 the caption is right but the figures are wrong (they are sitting postures) and the other way around for Figure 3, where they are the standing postures instead of the sitting ones.

Thank you for pointing this out. We have switched these figures now.